# Hazard Flagging as a Risk Mitigation Strategy for Violence against Emergency Medical Services

**DOI:** 10.3390/healthcare12090909

**Published:** 2024-04-27

**Authors:** Justin Mausz, Dan Piquette, Robert Bradford, Mandy Johnston, Alan M. Batt, Elizabeth A. Donnelly

**Affiliations:** 1Peel Regional Paramedic Services, 1600 Bovaird Drive East, Brampton, ON L6V 4R5, Canada; dan.piquette@peelregion.ca (D.P.); mandy.johnston@peelregion.ca (M.J.); 2Department of Family and Community Medicine, Temerty Faculty of Medicine, University of Toronto, 500 University Avenue, Toronto, ON M5G 1V7, Canada; 3Schulich School of Medicine and Dentistry, Western University, 1151 Richmond St, London, ON N6A 5C1, Canada; rbradford2025@meds.uwo.ca; 4Faculty of Health Sciences, Queen’s University, 99 University Avenue, Kingston, ON K7L 3N6, Canada; alan.batt@queensu.ca; 5Department of Paramedicine, Monash University, Building H, Peninsula Campus, 47-49 Moorooduc Hwy, Frankston, VIC 3199, Australia; 6School of Social Work, University of Windsor, 167 Ferry Street, Room 213, Windsor, ON N9A 0C5, Canada; donnelly@uwindsor.ca

**Keywords:** paramedics, emergency medical services, violence, workplace violence, occupational health and safety

## Abstract

Paramedics are increasingly being subjected to violence, creating the potential for significant physical and psychological harm. Where a patient has a history of violent behavior, hazard flags—applied either to the individual, their residential address, or phone number—can alert paramedics to the possibility of violence, potentially reducing the risk of injury. Leveraging a novel violence reporting process embedded in the electronic patient care record, we reviewed violence reports filed over a thirteen-month period since its inception in February 2021 to assess the effectiveness of hazard flagging as a potential risk mitigation strategy. Upon reviewing a report, paramedic supervisors can generate a hazard flag if recurrent violent behavior from the patient is anticipated. In all, 502 violence reports were filed, for which paramedic supervisors generated hazard flags in 20% of cases (n = 99). In general, cases were not flagged either because the incident occurred at a location not amenable to flagging or because the supervisors felt that a hazard flag was not warranted based on the details in the report. Hazard flagging was associated with an increased risk of violence during subsequent paramedic attendance (Odds Ratio [OR] 6.21, *p* < 0.001). Nevertheless, the process appears to reliably identify persons who may be violent towards paramedics.

## 1. Introduction

Violence against paramedics has been characterized as a “serious public health problem” [1], with the potential for significant physical [2] and psychological [3] harm. In a recent 10-year review of the United States (US) Bureau of Labor Statistics, emergency medical services personnel were found to experience a risk of lost-time injury from violence approximately six times greater than the US population and 60% higher than comparable health professions, such as nursing [4]. Given high rates of trauma exposure among paramedics [5], violence has the potential to compound what is already widely recognized as a mental health crisis within the profession [6], with downstream implications for community safety from workforce disruptions. There is, then, a compelling need to develop evidence-informed risk mitigation strategies that balance service delivery with paramedic safety.

One potential solution may be to “flag” individuals with a history of violent behavior such that paramedics can be alerted ahead of time if responding to a patient known to have an increased risk of violence. Our service developed a novel, point-of-event violence reporting process embedded in the electronic Patient Care Record (ePCR) [7] alongside a suite of new violence prevention policies. Paramedics are encouraged under the policy to complete a violence report immediately following the 9-1-1 call if they encounter an abusive, threatening, or assaultive person during the call. Upon receipt of a violence report, supervisors have the option to flag a patient’s residential address if there is a reasonably foreseeable risk of recurrent violent behavior. Hazard flags are read by dispatch when paramedics attend the residence during subsequent 9-1-1 calls, enabling crews to respond more cautiously or coordinate their response with police or other emergency services—potentially reducing the risk of harm from violence.

Therefore, as part of our program evaluation, our objectives were to (1) assess the proportion of cases eligible for flagging; (2) explore reasons why cases were not flagged; and (3) evaluate the impact of hazard flags on the risk of violence during subsequent paramedic attendances.

## 2. Methods

### 2.1. Study Design

We retrospectively reviewed data from Ambulance Call Reports (ACRs) and violence reports generated over a 13-month period since introducing a new violence reporting process into our ePCR system. We used patient-level identifiers to track violence reports that resulted in an address being flagged and compared patients with and without hazard flags for recurrent violent behavior.

### 2.2. Setting

Our work is situated in the Region of Peel in Ontario, Canada. Peel Regional Paramedic Services is a large, publicly funded paramedic service solely responsible for a population of 1.5 million residents across a mixed urban/rural geography of 1200 km^2^. At the time of this study, the service employed approximately 750 paramedics and responded to an average of 130,000 emergency calls per year, positioning the service as the second largest in the Province of Ontario.

### 2.3. The External Violence Incident Report

The development of the External Violence Incident Report (EVIR) is described in detail in an earlier publication [7]. Briefly, its development involved an extensive stakeholder consultation and pilot testing process to develop a streamlined, user-friendly incident report that is purpose-built to collect comprehensive information about violent encounters. Paramedics are prompted by ePCR software (https://www.interdev.ca/solutions/mobile-epcr-software/, accessed on 15 January 2024) to complete a report if they experience violence (i.e., verbal abuse, threats, sexual harassment, or physical or sexual assault) during a call. The form uses a combination of drop-down menu selections and checkboxes to gather quantitative data about violent incidents and incorporates a free-text box where paramedics can type a detailed narrative description of the incident. The form is available in the Appendix A.

### 2.4. Hazard Flagging

Paramedic supervisors are required to review completed EVIRs within 24 h of submission. Upon reviewing an EVIR, supervisors can generate a hazard flag for violence attached to a patient’s residential address if recurrent violent behavior is anticipated. The criteria for generating a hazard flag are articulated in a service health and safety policy. Examples include suspected criminal activity or the presence of persons with a known history of violence or who are exhibiting behavior that could potentially escalate into assaultive actions. In the management review section of the EVIR, supervisors are asked whether they generated a hazard flag, and if not, why not—with a free-text box in which they are to document their rationale. Hazard flags are maintained in a living database for a one-year period, after which the flag expires. As a policy, hazard flagging pre-dates the introduction of the new reporting process by several years, but the advent of enhanced surveillance vis a vis the EVIR allows for a more robust evaluation of its potential effectiveness.

The flags themselves do not prescribe a response protocol per se; rather, the paramedic crew uses the information from the flag in conjunction with the dispatch details of the 9-1-1 call to adapt their response accordingly. This may include, for example, having a supervisor, additional paramedic crews, or police attend the call.

### 2.5. Data Collection

Our study spanned a 13-month period following the launch of the new reporting process on 1 February 2021 through 28 February 2022. We abstracted all ACRs and EVIRs generated during the study period and substituted an alphanumeric identification code for the patient’s name. This information was assembled into a dataset that included the patient ID code, the total number of 9-1-1 calls attended, and the number and type of EVIRs produced for each patient during the study period. Where a hazard flag was generated in response to an EVIR for a patient, we documented the sequential call number on which the flag was placed (i.e., first, second, third, etc.) and the number of violence reports filed.

### 2.6. Measures/Outcomes

Although hazard flags are applied to a residential address, our unit of analysis for recurrent violent behavior was necessarily constrained to the patient ID code, given that multiple people can reside at the same address. Our primary outcome was whether there were documented instances of recurrent violent behavior attributable to a patient for whom a hazard flag had been generated.

### 2.7. Analysis

We analyzed our data in three ways. First, we used descriptive statistics to report on and characterize the proportion of violence reports resulting in hazard flags being generated (Objective 1). Next, we thematically reviewed supervisor notes using qualitative content analysis [8] to identify common reasons why hazard flags were not assigned (Objective 2). Finally, we used chi-square tests to estimate the probability of violence during repeat paramedic attendance for flagged vs. not flagged patients (our primary outcome; Objective 3) where complete data was available.

## 3. Results

Our results are summarized in Figure 1. A total of 722 paramedics responded to 119,683 emergency calls, for which 271 paramedics filed 502 EVIRs, with 37% of reports (n = 187) documenting some form of physical or sexual assault and 9% (n = 45) resulting in the paramedic being physically harmed. This corresponded to 37% of the active-duty paramedic workforce experiencing violence, 19% being assaulted, and 6% sustaining physical injuries after an assault during the study period, with an average of one assault every 3 days.

### 3.1. Objective 1: Distribution and Characteristics of Hazard Flags

Among violence reports, 20% (n = 99) resulted in a hazard flag being generated. On average, hazard flags were generated on the second call (±4.37) and the first report of violence (±0.67). In eight cases, the report indicated an existing hazard flag was not communicated by dispatch. Our point estimates suggested that reports documenting a physical or sexual assault (Odds Ratio [OR] 0.95) or injurious assault (OR 0.48) may be less likely to result in a hazard flag, but neither estimate met significance at the 5% threshold (*p*-values 0.838 and 0.128, respectively). While 50% (n = 95) of cases involving a physical or sexual assault occurred at a private residence, the remainder took place at other locations not amenable to flagging, most commonly on a street or at an intersection (20%; n = 38), long-term care home (9%; n = 17), hotel (4%; n = 7), or store/restaurant (5%; n = 10). Cases involving an injurious assault were less likely to occur at a private residence amenable to flagging under the policy (OR 0.32, 95% Confidence Interval [CI] 0.16–0.62, *p* < 0.001)

### 3.2. Objective 2: Reasons for Not Generating Hazard Flags

Information regarding hazard flag creation was missing from supervisor documentation in 13% (n = 67) of cases. In the 67% (n = 336) of cases where a hazard flag was not created, supervisors indicated that a flag already existed (5%; n = 19) or the incident occurred at a non-residential address (35%; n = 118); however, in most cases (60%; n = 199), supervisors selected “other” as the reason for not generating a hazard flag, providing a rationale in their typed notes.

Our thematic analysis of supervisor notes is presented in Table 1. Most commonly, the supervisor explained that the incident occurred at a location not amenable to flagging under current policy (such as a congregate living facility or public place) or that the perpetrator did not have a fixed residential address that could be flagged. Naturally, the decision of whether to generate a hazard flag carries with it an element of subjective judgment on the part of the supervisors, and we observed that the supervisors appeared to apply a framework of severity, “volitionality”, and verifiability in determining whether a violent incident warranted a hazard flag. Some examples are provided below.

“The family members’ behavior, while verbally assaultive and intimidating, does not warrant (a) hazard flag at this time”. “While the patient has displayed racist and horrible behavior, unsure it warrants a flag at this time”. “Patient was under the influence of alcohol/drugs, which could have been a one-time incident”. “While (the) sexual assault attempt is very real, not able to confirm, as the patient denied it and stated otherwise”.

### 3.3. Objective 3: Effectiveness of Hazard Flags

Complete patient and hazard flag data was available for cross-tabulation in 413 cases (82% of the sample). When a hazard flag was generated, the risk of repeat violence perpetrated by the same patient was six times higher than when a hazard flag was not generated (OR 6.21, 95% CI 2.29–16.87, *p* < 0.001). Most patients (89%) were associated with only one EVIR (mean 1.16 ± 0.70); however, one patient was documented in twelve reports—more than double that of the next most frequently documented patient (with five reports). After excluding this patient, the mean number of violence reports per patient decreased to 1.14 (±0.46), and the OR for recurrent violence given a hazard flag dropped to 5.59 (95% CI 2.01–15.50, *p* < 0.001).

## 4. Discussion

Our goals in this study were to evaluate the effectiveness of hazard flagging on the risk of recurrent violent behavior towards paramedics in a single service in Ontario, Canada. We found that only a minority (20%) of violence reports resulted in a hazard flag being created, that more serious (i.e., injurious assaults) were somewhat less likely to be flagged, and that hazard flagging was associated with a substantially increased risk of recurrent violence during subsequent 9-1-1 calls for the same patient. On the face of it, our results paint a discouraging picture of the potential of hazard flagging as a risk mitigation strategy, but there is nuance worth unpacking.

First, it is worth noting that when a hazard flag was created, there were no further reports of violence in 88% of subsequent paramedic attendances for the same patient. While the increase in relative risk is striking, the absolute increase is from 0.005% to 0.008% of all 9-1-1 calls for not flagged vs. flagged patients, respectively. Cases where patients were documented in two or more violence reports (n = 45) made up 9% of the total sample of EVIRs. Incomplete data obscure this picture somewhat, but weighted against the overall call volume, the prevalence of documented incidents of recurrent violent behavior from individual patients is small. Another interpretation is that the hazard flagging system consistently identifies persons with a history of violence who present an increased risk of violent behavior during encounters with paramedics. While we saw additional violence reports filed after a flag was generated, it may be that the flag prompted the paramedics to modify their response strategy to reduce the risk of injury during the encounter. Unfortunately, our preliminary data did not allow for a detailed examination of the risk of injury from recurrent violence, but this would be a useful line of inquiry in future work.

From a policy perspective, hazard flagging as a violence risk mitigation strategy is an attractive concept with a sound premise: when a person is known to have a history of violent behavior, conveying that information to the responding paramedics has the potential to strengthen not just provider safety but patient safety as well. In our system, hazard flags for violence lack granularity on the nature and type of risk, but such a system could be optimized to grade the risk of recurrence and the potential for harm in a “violence risk matrix”. Violence risk assessment tools are widely and successfully used in healthcare settings [9], and incorporating a violence risk assessment in a hazard flagging system could enable coordinated, interagency response plans tailored to both the nature of the risk and the level of potential harm. A cognitively impaired patient in a long-term care home with a history of biting staff (for example) requires a different response strategy than a person with complex mental health needs who has access to weapons and is threatening “suicide by cop”—both cases from our dataset. Both patient and provider safety need to be considered in organizing an optimal response.

One important limitation we observed in our current process is the trend towards more serious incidents being less amenable to flagging under current policy. This was largely because injurious physical assaults happened more commonly in public settings not amenable to flagging or were perpetrated by patients without a fixed residential address. Person-level flags may be more attractive but are harder to implement in practice if obtaining patient names is not part of the call-taking process in the communications center. Alternatively, attaching hazard flags to phone numbers may be more fruitful, given the ubiquity of mobile phone use in North America, but obviously, care must be taken not to inadvertently flag individuals who did not perpetrate the violence themselves (e.g., if a third party calling on behalf of a patient).

As an occupational health issue, violence against paramedics demands our attention. In our sample, more than a third of our active-duty paramedic workforce reported exposure to violence, with nearly 20% experiencing an assault and 6% sustaining a physical injury. Our work parallels a growing body of research illustrating that paramedics are at substantial risk for both physical injury and psychological sequelae [3,4,10] from violence, including post-traumatic stress disorder, burnout, and exit from the profession. Particularly in Canada, there has been an “exodus” of physicians, nurses, and allied health personnel leaving healthcare [11,12]—a problem worsened by increasing exposure to workplace violence [13] that likely extends to the paramedic context as well. This creates an important vulnerability in community safety if violence is contributing to paramedic staffing shortages.

### Limitations

Our work should be interpreted within the context of certain limitations. First, we note with concern the large proportion of reports with missing or incomplete hazard flag information. This complicates our assessment of the potential effectiveness of hazard flagging as a risk mitigation strategy and illustrates the importance of supporting supervisory staff in actioning violence reports after submission. In response to these findings, our service implemented several changes to the hazard flagging process and provided additional education to supervisors on the goals of the program as a risk mitigation strategy and the importance of contributing to a supportive workplace culture. We are in the process of evaluating the results. Second, our program was developed within our specific legislative and policy context—readers should take care when extrapolating our findings or adopting our processes in their own regions. Third, limits on the scope, type, and availability of data mean we are unable to comment on several important evaluative questions, including the degree of harm the paramedics experienced, the ways in which they may have adapted their response to patients with a known history of violence, and the degree to which the latter may have affected the former. Future research on these points would be welcome.

## 5. Conclusions

As a concept, hazard flagging for violent patients shows promise as a potential risk mitigation strategy. While we did not observe a decrease in recurrent violence from flagged patients, the process appears to identify persons who present an increased risk to paramedics. In that respect, there is ample opportunity to refine the process to improve data collection, capture a larger proportion of violent patients, and stratify paramedic response according to a more granular risk assessment.

## Figures and Tables

**Figure 1 healthcare-12-00909-f001:**
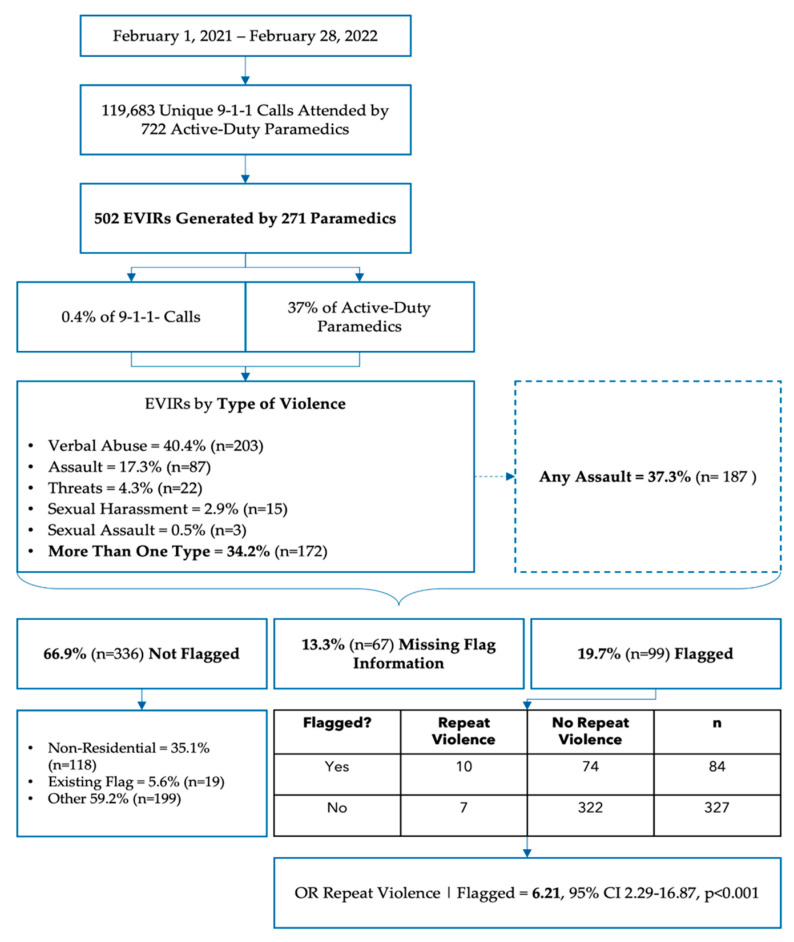
Results overview. EVIR = External Violence Incident Report; OR = Odds Ratio.

**Table 1 healthcare-12-00909-t001:** Thematic analysis of supervisor notes that document reasons why a hazard flag was not generated. Note that counts may sum to more than the number of cases in the study sample as the supervisors sometimes gave more than one rationale for not generating a flag for a particular report.

Reason	Count	Explanation	Examples
Not a Flaggable Address	74	Situations in which the call location is not a residential address that could be flagged under the policy.	“Pt. (patient) is NFA (no fixed addressed.)” “Public area”. “Patient picked up outside a restaurant”. “Subject does not live at pick-up address”. “Group home”. “CNO (could not obtain) address”.
“Not Required”	66	Supervisors’ notes that indicate (after reviewing the report) that a hazard flag is “not required”, without explaining why.	“N/A” “Not required” “Not appropriate” “Not necessary”
Low Risk	36	An assessment that the nature of the incident does not pose a sufficiently serious risk from violence to warrant the creation of an address hazard flag.	“Nature of call would naturally trigger (police attendance). Type of call is mental health and per crew, no injury. Furthermore, overuse of flagging may be counterproductive”. “Likelihood of repeated calls may be moderate to high, however physical risk to paramedics may be considered low”. “No physical threat of violence”. “The family members behavior, while verbally assaultive and intimidating, does not warrant a hazard flag”. “No direct assault on paramedics”. “While patient has displayed racist and horrible behavior, unsure that it warrants a flag”.
Not Volitional	26	An assessment that the violence documented in the report was not intended to cause harm, and an address hazard flag was not warranted as a result.	“Patient is cognitively impaired”. “Type of occurrence is mental health per crew”. “Reversible medical cause of aggressive state”. “Patient was under the influence of alcohol/drugs, which could have been a one time incident”. “Dementia patient” “Post-ictal, behavior due to medical condition”. “Medical issue”. “Patient was intoxicated at the time”. “Patient was under the influence of alcohol” “Patient is impaired”.
Insufficient Detail	12	Notes indicating that the violence report did not contain enough information to justify the creation of an address hazard flag.	“Not enough detail provided by paramedics to create a hazard flag for verbal abuse and threats of violence”. “Not enough info”. “Not enough detail provided to warrant a flag”.

## Data Availability

Data for this study may be shared with interested researchers on a case-by-case basis, subject to a privacy review and formal data sharing agreement.

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
