# Peer review of "Hazard Flagging as a Risk Mitigation Strategy for Violence against Emergency Medical Services"

_healthcare, 2024, doi:10.3390/healthcare12090909_

Round 1
Reviewer 1 Report
Comments and Suggestions for Authors
I have the following comments:
- Does including verbal abuses and threats in risk reporting lead to over-reporting?
- The process of flagging high-risk cases needs an objective basis, such as pre-established SOPs or criteria. Only based on the supervisors' personal opinion, it is not only too subjective, consumes working time, and is not cost-effective. An information system should be used instead.
- The patient's rights will be affected once an address is flagged as high-risk. So, the supervisor should list the reasons for the high-risk flag rather than the reasons for the unflagged patient. In addition, the number of patients flagged is small, and the supervisor is more willing to analyze in detail and provide feedback/education to the team members.
- The biggest drawback of this system is that it cannot flag cases without a fixed address or cases that occurred in public spaces. Although the above two types of cases may be highly risky, this system cannot cover them.
- If the supervisors' evaluation and feedback cannot be strengthened, the system will likely continue increasing the number of flagged addresses.
- The C.I. is too wide, indicating that too few cases are analyzed and the statistical results are unreliable.
Author Response
Dear colleague,
Thank you for taking the time to review our manuscript. Your observations are helpful in clarifying our thinking on this topic ansd we have incorporated several edits in response to your comments. Our work is strengthened as a result. Please refer to the enclosed table for more specific information.
Kindly,
Justin

Reviewer 2 Report
Comments and Suggestions for Authors
The subject title is of interest. However, it is difficult to design a research method on this subject due to both physical and psychological forms of violence. Nevertheless, the ideas of the researchers and the article are read with interest. It is well explained and the language is clear enough.
I have no additional suggestions regarding the article.
Author Response
Dear colleague,
Thank you for taking the time to review our manuscript. We agree this is a complex and challenging topic to research; the present program evaluation study is part of a larger, multi-phase and mixed methods program of research broadly intended to generate important epidemiological research on this issue. We are working diligently behind the scenes to apply the lessons we have learnt from the various evaluation strategies - including the enclosed study on hazard flagging - to mitigate the risk of harm to our paramedics. We are grateful for your contribution in reviewing our paper and hope you will keep up to date on our work as additional papers are published.
Kindly,
Justin
Reviewer 3 Report
Comments and Suggestions for Authors
A great paper--and a very important one. Here are some recommendations and/or reflections I have from your paper:
There were some phrases or sentences that were not clear to me.
Under 2.3, what do you mean by “derivation phase?”
Under 2.4, this sentence is confusing: “Paramedic-supervisors action completed EVIRs within 24 hours of submission.”
I may have missed rationale for a few things:
1. Why can’t hazard flags be created for individuals versus residences? (section 2.6)
2. Why can’t hazard flags be done for non-residential locations-if there is a drop-down menu as part of the electronic record? (Objective 1)
3. Do supervisors have to fill out the forms? Why not the paramedics? (Objective 2)
On page 5, I was really surprised by some of the supervisor statements. It seems that it would be better to err on the side of being more conservative and flagging anything with potential harm. For example, an individual may cause violence in a non-volitional manner, but if they’ve committed any harm, it seems that they should be flagged. Is it possible for you to add something about culture in your Discussion—to indicate the hesitancy of supervisors to flag someone who has been violent, even if there may be an explanation for their behavior? Do you think it is cultural?
Author Response
Dear colleague,
Thank you for taking the time to review our manuscript and for your encouraging feedback. Your comments have helpful in clarifying several points in our manuscript and our work is strengthened as a result. Please see the enclosed table for a more detailed response to each of the points you raised and accept our gratitude for your time.
Kindly,
Justin
